# *Domesticating* Colour in the Early Modern Age: Dyeing Wool in Black in Portugal

Luís Gonçalves Ferreira 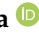

Lab2PT-IN2PAST, Institute of Social Sciences, University of Minho, 4710-057 Braga, Portugal; id9365@alunos.uminho.pt

**Abstract:** Mastering a colour—as such, its 'domestication'—involves a weft of technological and symbolic relationships encompassed in the human ability to reproduce a visible colour using the techniques of textile dyeing. The *Regimento dos panos* or *Regimento dos trapeiros* ('regulation of fabrics' or 'regulation of drapers'), published in 1573 and expanded in 1690, is a document made up of 107 chapters aiming to standardise the various stages of the production chain of woollen goods in Portugal. In the sections relating to the finishing of fabrics, the regulation carefully details the dyeing of the colour black. The main aim of this text is to discuss the four recipes presented in that document. The system presupposed a phase exogenous to the rules, since the fabrics had to be previously dyed blue ('celestial blues') by means of successive immersions of the cloth in a vat with indigo. The dyeing itself was achieved by mixing mordants and auxiliaries (alum, tartar, iron sulphate, and tannins) with a red dye (madder). The main conclusion is that the formulae presented do not constitute, in their general principles, a characteristic Portuguese methodology. In addition, the article includes an inventory of the raw materials used for dyeing in the Early Modern Age, produced, through a qualitative method, through cross-reference with other manuscript and printed sources, as well as an interpretation of their social and economic importance, and a systematisation of the types of Portuguese wools.

**Keywords:** black; blue; early modern age; natural dyeing; Portuguese wools





## 1. Introduction

Colours are omnipresent in human daily life, but it is not possible to precisely date their 'domestication'. This verb, when applied to biology and the relationship between human beings and the other species around them, determines interdependence between the domesticated and the domesticators. Domesticated species differ from wild species in that they depend on human beings to develop in a co-evolutionary environment arising from ecological interaction based on mutualism [1] (pp. 663–665). The problem is more complex than the simple survival of a species thanks to its interaction with humans. In the context of this work, 'domesticating a colour' involves a weft of technological and symbolic relationships encompassed in the human ability to reproduce a visible colour using the techniques of textile dyeing. Domestication seems to have occurred at different stages throughout history, working through the three dimensions of colour: 'hue (the name), intensity (brightness/dullness), and value (lightness/darkness)' [2] (p. 280).

It is difficult to determine what led the first hominids to understand that their colours and those of the plant and animal world could be reproduced using fauna, flora, or fungi. When did human beings first colour their bodies? When did they paint objects? When did they discover that they could apply colours to the skins and fabrics they wore? When was water first mixed with dyes to create a dye bath? When were mordants added? In what context were the techniques for making insoluble substances soluble-mastered? These questions contain the different complexities of the human relationships with colours and the various dyeing technologies. In these terms, the domestication of colour is a process

that dates back several millennia, long before the emergence of writing, and is therefore vulnerable to the conditions of the survival of material culture. As evidenced by wall paintings over 30,000 years old, human beings were already using pigments to express themselves artistically before they mastered dyes [3] (pp. 19–20). The field of study remains open to new archaeological discoveries that may fill in the blanks left by the documentary silences. However, there are two fundamental premises relevant to the present work: the phenomenon of the domestication of colours, from painting to dyeing, cutting across the world's civilisations; even if their designation remained the same, the social prestige of colours changed with the variables of history (time and space). In the case of textiles, the systems of socio-cultural apprehension and hierarchisation depended on technical developments in dyeing [4] (pp. 76–80).

The *Regimento da Fábrica dos Panos destes Reinos*, published on 7 January 1690, consists of 107 chapters aimed at standardising the production chain of woollen goods in Portugal. This document, issued by King Pedro II (1648-r.1683-1706), added to another, of 96 chapters, published in 1573 by King Sebastião (1554-r.1557-1578). Given the economic importance of woollen cloths, the Portuguese monarchs of the Early Modern Age regulated one of the fundamental sectors of the Portuguese economy by establishing technical requirements that guaranteed the quality of cloth production. The mercantile policy of the last quarter of the seventeenth century—in which context interest in woollen cloths was revitalised—sought to protect the kingdom's production and ensure Portuguese manufactures the conditions to compete with their foreign counterparts. This regulation was a fundamental element in the development of the wool industry in Portugal [5] (p. 3).

This article sets out one primary objective and three secondary goals: to discuss the recipes for black dye as presented by the regulation; to systematise the hierarchy of wool quality based on the stages of production; to create an inventory of the main dyes and mordants used in Portugal during the Early Modern Age; and to investigate the social and economic values of these products within the Kingdom and in the Empire. To answer these questions, the presentation will be divided into three parts. The first concerns sources and methodology. The second part sets the regulation of fabrics within the context of Portugal in the second half of the sixteenth century and the mercantile policy of the end of the following century. The last part discusses the recipes proposed for dyeing the colour black in two stages: firstly, the blue standard, obtained through several immersions of the cloth in an indigo bath and verified by crown-appointed officials; secondly, the actual dyeing of black, whose ingredients varied depending on the indigo and the desired colour saturation, achieved through baths with mordants (metallic salts, tannins, and tartar), colour darkening/saturating agents, and red dye (madder).

## 2. Sources and Methodology

Although some recipes can be found in sources from the High Middle Ages, it was between the fourteenth and sixteenth centuries that dyeing manuals proliferated in the main European countries. Some documents produced in the territories of the Holy Roman Empire, the Italian Peninsula, Flanders, and the Iberian Peninsula (Valencia, in present-day Spain) stand out [6] (pp. 25–34). Unlike other European regions, no textile dyeing manuals produced in Portugal or written by the Portuguese during the Early Modern Age have yet been identified[1]. Scientific studies from the last twenty years approaching the issue from a historical perspective are limited[2]. The cloth regulation that forms the empirical basis of the present analysis carefully details the dyeing of black. This allows an insight into the formula proposed for making the colour and, consequently, experimental approaches to organic dyeing. The main aim of this analysis is to discuss the dyeing of black according to the recipes presented in the fabric regulation. Analysing this document was a paradigmatic opportunity for this purpose. The cross-referencing of sources allowed for an inventory of the dyeing plants, animals, and fungi documented between the sixteenth and eighteenth centuries to compiled and for their social values to be approached through the prices of some of these raw materials. An endeavour was also made toward systematising the

classification and hierarchisation of Portuguese wools according to the amount of wool used and the fineness of the warp yarns, as well as discussing the role of municipalities in overseeing the quality of the fabrics and of the blue dyeing.

In addition to the mentioned regulation, other manuscript and printed sources were analysed: the Valentim Fernandes Codex (early sixteenth century), the *Regimento do pastel* ('regulation of dyer's woad') (1536)[3], the *Regimento dos Tintureiros de Lisbon* ('regulation of the dyers of Lisbon') (1572)[4], the *Descrição do Reino de Portugal* ('Description of the Kingdom of Portugal') (1610) by Duarte Nunes do Leão [11], the *Vocabulário Português e Latino* ('Portuguese and Latin Vocabulary') (1712–1720) by Rafael Bluteau, and a customs tariff from 1752 [12][5]. A mixed methodology was used to work the documentary corpus[6]. The date generally pointing to the end of the Early Modern Age was pulled back a little, taking into consideration changes in the standards of textile production and consumption (increased importance of cotton[7] and a resurgence of wool) and by the emergence of synthetic dyes (1856)[8]. It is this author's opinion that the confluence of these events changed the socio-economic structures dictating the effectiveness of the sources used, which considered the manufacturing of organically and manually dyed wools. The critical documentary analysis was undertaken with a view to contribute to a historiography of colour that goes beyond the scientific sphere onto the cultural and artistic realms. Ultimately, an effort was made to make the data collected available in such a way that they can be easily transferred to the workshops of craftsmen and artists as laboratory tools[9].

### 3. The Regulation of Fabric

The *Regimento da fábrica dos panos* ('regulation of fabric mills') or *Regimento dos trapeiros* ('regulation of drapers') by King Sebastião, published in 1573, has not received much attention in Portuguese historiography[10]. The context that led to its expansion and republication in the following century is better known. In the preamble to the 1690 text, King Pedro II clarifies the reasons for the regulation: a law had been published banning the use of foreign fabrics, and it was therefore urgent to reorganise and protect domestic production. The greatest problems with the manufacturing of woollen goods were to be found in weaving and dyeing, as non-compliance with the existing regulations resulted in 'poorly woven fabrics, fraudulent both in yarn count and widths and in the impropriety of the dyes [used]' [15] (p. 3). At the beginning of the nineteenth century, wool still accounted for 80% of consumption in the world's textile market, followed by linen and cotton [16] (p. 308). Although this consumption pattern gradually changed over the course of the nineteenth century[11], wools were central to the daily lives of people from all social strata during the Ancien Régime, in the form of fine or coarse fabrics, either dyed or in the natural colours of the fibres. Their uses were diverse and included clothing, protective clothes against the rain and cold, hats, blankets, and other artefacts such as tapestries, *armações* (textile furnishings), and other forms of textile architecture, all of which aimed to create comfort, deceive the senses, and convey lustre or grandeur to interiors, which had less diverse furniture than is common nowadays.

The reissuing of the regulation and the publication of the law banning foreign fabrics [17] during the last quarter of the seventeenth century should be interpreted in the light of protectionism. This was a structural sector in the Portuguese economy and accounted for the largest percentage of consumption intended to fulfil basic and superfluous needs. Inspired by French doctrine and concerned with the negative trade balance in Portugal, the state's economic policies sought to promote manufactures that were strategic for public finances, such as silk, glass, blacksmithery, and woollen products. The efforts to reorganise these industries were made in areas historically linked to these activities. In the case of woollen fabrics, these were the region of Beira Interior (an inland northern region of Portugal), especially near the mountainous regions of Serra da Estrela and Serra de Montejunto, and that of Alentejo. The area of Serra da Estrela combined the availability of raw materials, thanks to the transhumance of sheep between the plains and the mountains, with the abundance of water, irrigation being essential in manufacturing and at various other stages

of the production and finishing of cloths [18] (pp. 180–190). The wool-production centre of Beira Interior, with its epicentre in the town of Covilhã, was the protagonist of the three manufacturing surges of the Portuguese Early Modern Age [19] (pp. 232–263).

Going back to the second half of the sixteenth century, the specific political and economic motivations that led to the publication of the regulation by King Sebastião are not known. The intentions behind the establishment of a system to supervise production, based on the municipal elites and the power of the *vedores dos panos* ('overseers of cloths'), and to control the various stages of the wool production chain still await inquiry. The first few years of King Sebastião's personal reign were marked by intense legislative activity with his intent focused on reform. He invested in the military reorganisation of the kingdom (establishing the ordinance system), in coastal defence, in the organisation of the naval system to ensure the safety of Portuguese maritime trade, in the regulation of the minting of silver coins, in the improvement of the application of justice, and in customs moralisation and reform, which included a law to curb spending on silks and other sumptuary goods [20] (pp. 194–219). Still during the 1570s, in Lisbon, the regulations of the city's mechanical officials were compiled and reformed by Duarte Nunes do Leão; legislation aimed at standardising weights and measures was published in 1575. The system of *palmo craveiro*[12] was instituted as a metrological basis for length, including for textiles; it would prevail until the adoption of the decimal metric system in 1852.

This reformist drive is justification enough for the monarch's will to act on textiles in general and woollen goods in particular. In Portugal, as in the rest of Europe, this was the manufacturing sector that occupied most of the labour force. Fabric production—an activity deeply rooted in domestic production and that responded to a basic need—was spread throughout the country, depending on the availability and specificities of the raw materials. Since the end of the Middle Ages, in parallel with manufacturing for self-consumption, production centres emerged with density of production and marked regionalisation of consumer markets. Flax and linen production was mainly concentrated on the coast, especially in the region of Entre Douro e Minho. Sericulture developed in the Trás-os-Montes region and in certain urban centres such as Évora, Lamego, Lisbon, and Porto. Cotton—a secondary fibre until the eighteenth century, as it was not grown in the kingdom—accumulated capital and labour in the regions with a tradition of linen production (Lamego and Tomar). As mentioned, the wool industry was scattered across the inland strip stretching from the north of Beira Baixa to Baixo Alentejo. Some urban workshops gathered the whole by articulating agricultural and industrial productions [18] (pp. 95–100). As Table 1 illustrates, certain manufacturing centres showed levels of specialisation in terms of the quality of the fabrics produced. The fabric standardisation and inspection system established by the *Regimento dos panos* recognises the fundamental role of the urban workshops of Covilhã, Portalegre, and Estremoz in the manufacturing of woollen fabrics (Table 2).

**Table 1.** Characterisation of wool production centres by warp density, 1720.

| Production Centre (a) | Correspondence in the Regulation (a) | Number of Wool Yarns in Warp (b) |
|---|---|---|
| **Estremoz, Arronches, Vila Viçosa, and Monforte** | *Dozeno* | 1200 |
| **Castelo de Vide and Elvas** | *Dezocheno* | 1800 |
| Manteigas | *Vinteno* | 2000 |
| Covilhã | *Vintedozeno* | 2200 |

Sources: (a) [22] (p. 224); (b) [15] (pp. 7–11).

**Table 2.** Wool producing centres involved in the triennial renovation of blue standards, 1573/1690.

| Locality | Officials Involved |
|---|---|
| Covilhã | *Corregedor da comarca* ('district judge'), *vereadores* ('councillors'), *procurador do concelho* ('county procurator'), and six officials (two drapers or two dyers from Covilhã, Portalegre, and Estremoz) |
| Portalegre | *Corregedor da comarca* and officials (drapers or dyers) from Covilhã, Portalegre, and Estremoz |
| Estremoz | *Corregedor da comarca* and six officials (drapers or dyers) from Covilhã, Portalegre, and Estremoz |

Source: [15] (pp. 39–40).

One of the hallmarks of the European manufactures of this period was the presence of highly specialised merchants or officials who coordinated domestic production. These professionals combined rural industries, responsible for the initial stages of manufacture (spinning and weaving), with urban industries, centred on more complex tasks such as improving/finishing fabrics (dyeing or printing) [18] (pp. 96–97). They operated in an economic environ marked by an intertwining of agriculture and industry, the latter being a secondary activity to the former when carried out between agricultural tasks. It should be kept in mind that, in this period, the majority of the Portuguese population lived in rural areas. In the context of this regime, it was the draper who took on the role of mediating the various stages of the wool production chain.

The king entrusted certain representatives of the municipalities with the execution of the blue standards ('celestial blues'). Every three years, on a rotating basis, the judge, councillors, county procurator and six officials had to review this system, which organised the fabric regulation (see Table 2). The *vedores* ('overseers') of the fabrics were responsible for supervising production and imposing fines on offenders. The *corregedores da comarca* were magistrates appointed by the king every three years and were part of the crown's peripheral administration [23] (pp. 199–206). These royal officials had the right of visitation over the houses of the weavers, drapers, carders, and fullers [15] (p. 46). The reform of the regulation by King Pedro II added a new supervisory figure: the *juiz conservador*, who was inherently the *juiz de fora*, a judge, external to the council and designated by the king [15] (p. 47). These individuals were thus central to the articulation between the central administration, close to the crown, and the peripheral administration [23] (pp. 196–199). By appeal and interlocution, the *juiz conservador* was to become acquainted with the verdicts and decisions of the *vedores dos panos* and inspect the *vedor* and the fabric manufacturers annually.

As with other aspects of the administration, the system delegated supervisory and economic powers to the councils, divesting the central, regal power to those of the peripheries—the municipalities. Although the king had ultimate *auctorictas* (lit. authority) over his domains, he did not always have *potestas*, i.e., the ability to exercise the rights of sovereignty (collecting taxes, recruiting people for war, and executing justice). The Modern state was structured by delegating powers to the municipalities, particularly in the collection of taxes such as *sisas* (a tax calculated as an addition to the value of the sale of goods, as in modern Sales Tax or VAT), in the organisation of military ordinances, in the application of sanitary measures, and in certain matters of an economic nature such as the setting of fees by *almotacés* (inspector of weights and measures) or the tabling of prices [24] (pp. 162–165).

## 4. Black, Domesticated

After the wools had been sorted, washed, picked[13], carded, spun, and warped, they were woven (Table 3). The fabrics were categorised and marked according to the number of yarns in the warp (see Table 4). The thinner the threads, the higher the quality—the "*bondade*" (lit. goodness)—of the fabrics. The differentiation between coarse and fine fabrics was extremely important for the categorisation of textiles during the studied period. The

classification began by dividing the wool from intact fleeces[14]. This system had an impact on the transparency, comfort, lightness, brightness, and impermeability of the fabrics. The drapers were obliged to invest a tenth of their production each year in fine fabrics to protect the weavers' investments. The sizes of the combs varied according to the type of fabric, and there were specific, strict criteria for accumulating them[15]. As can be seen in Table 3, weaving was covered in 26 chapters of the regulation. This attests to the importance of this stage in the chain of value of wools.

**Table 3.** Stages of wool manufacturing, 1573[16].

| Step | Regulation (Chapters) | Chapters (Sum) |
|---|---|---|
| Sorting | I | 1 |
| Washing | II | 1 |
| Picking | III–IV | 2 |
| Carding | V | 1 |
| Spinning | VI | 1 |
| Warping the loom | VII | 1 |
| Weaving | VIII–XXXIV | 26 |
| Fulling | XXXV–L | 15 |
| Dyeing | LI–LXXIV | 23 |
| Shearing | LXXV–LXXVII | 2 |
| Inspection | LXXVIII–XCVI | 18 |

Source: [15].

**Table 4.** Classification and characteristics of woollen fabrics according to their weave, 1573/1690.

| Designation of the Cloth | Number of Yarns in Warp | Width of Comb (*Côvados*) | Width of Selvedge | Wool in Weave per *Ramo* (4) (*Arrátel*) | Wool in Warp per *Ramo* (*Arrátel*) | Weight of the Cloth per *Ramo* (*Arrátel*) | Mark |
|---|---|---|---|---|---|---|---|
| *Frisas* (1) | 730 | 2 2/3 | unknown | 3 | unknown | unknown | unknown |
| *Baeta dozena* (1) | 1200 | 2 7/8 | unknown | 3 1/2 | unknown | unknown | unknown |
| *Dozeno* | 1200 | 3 1/6 | 1 *sesma* | 3 | unknown | unknown | XIIB [mark of place] [weaver's mark] |
| *Quatorzeno* | 1400 | 3 1/3 | 16 or more yarns (on each side) | 3 1/2 | unknown | unknown | XIIII [mark of place] [weaver's mark] |
| *Quatorzeno dezimado* | [1400] | unknown | unknown | 2 | 4 | 6 | [caption (5)] |
| *Baeta sezena* (1) | 1600 | 3 1/8 | unknown | 4 | unknown | unknown | unknown |
| *Guardalete* | (at least) 1600 | 3 1/8 | unknown | 4 1/2 | unknown | unknown | unknown |
| *Pano de Cordão* (1) | (at least) 1600 | 3 1/8 | unknown | 4 1/2 | unknown | unknown | unknown |
| *Picote* (1) | (at least) 1600 | 3 1/8 | unknown | 4 1/2 | unknown | unknown | unknown |

**Table 4.** *Cont.*

| Designation of the Cloth | Number of Yarns in Warp | Width of Comb (*Côvados*) | Width of Selvedge | Wool in Weave per *Ramo* (4) (*Arrátel*) | Wool in Warp per *Ramo* (*Arrátel*) | Weight of the Cloth per *Ramo* (*Arrátel*) | Mark |
|---|---|---|---|---|---|---|---|
| *Sezeno* | 1600 | 3 1/2 | 18 or more yarns (on each side) | 3 3/4 | unknown | unknown | XBI [mark of place] [weaver's mark] |
| *Sezeno* [1] *dezimado* | 1600 | [caption (2)] | unknown | 2 1/4 | 4 | 6 1/4 | [caption (5)] |
| *Dezocheno* | 1800 | 3 3/4 | 12 *dobrados* (on each side) (3) | 4 | unknown | unknown | XBIII [mark of place] [weaver's mark] |
| *Dezocheno dezimado* | 1800 | [caption (2)] | unknown | 2 1/2 | 4 1/2 | 7 | *marcas são da* [caption (5)] |
| Vinteno | 2000 | 4 1/8 | 12 *dobrados* (on each side) | 4 1/4 | unknown | unknown | XX [mark of place] [weaver's mark] |
| *Vinteno dezimado* | 2000 | [caption (2)] | unknown | 2 3/4 | 5 | 7 3/4 | [caption (5)] |
| *Vintedozeno* | 2200 | 4 1/4 | 12 *dobrado* (on each side) | 4 1/2 | unknown | unknown | XXII [mark of place] [weaver's mark] |
| *Vintedozeno dezimado* | 2200 | [caption (2)] | unknown | 3 | 5 1/4 | 8 1/4 | [caption (5)] |
| *Vintequatreno* | 2400 | 4 1/2 | 12 *dobrados* (on each side) | 5 | unknown | unknown | XXIIII [marca do lugar] [weaver's mark] |
| *Vintequatreno dezimado* | 2400 | [caption (2)] | unknown | 3 1/4 | 5 3/4 | 9 | [caption (5)] |

[1] The document reads *dozeno*, but, given the number of yarns in the warp, it should instead be *sezeno*. Source: [15] (pp. 7–18). Captions: (1) These cloths were likely *gaspeados*. The source suggests it is a finish administered to these fabrics. (2) *Os pentes em que se tecer serão os próprios (...) dos panos verbis* ('the combs to be used to weave shall be the very ones (. . .) of the *verbis* cloths', chapter 28). (3) A *dobrado* is probably a twisted yarn. (4) Unit of measurement referring to the length of the warp, which, according to the *Regimento*, was the equivalent of 6 1/3 *côvados*. (Table A1). (5) *Marcas são da feição dos panos verbis, porque em lugar do B, que levará no pano verbi (...), levará o dezimado um D por onde se conheça* ('the marks are in the manner of the *verbis* cloths, although in the place of the B of the *verbi* cloth, the *dezimado* carries a D so it can be identified' (chapter 28).

The regulatory system distinguished between *dezimados* and *verbis* fabrics. The first had the weight of the wool fixed to the length, both for the warp (*levado a urdir*, lit., 'that it takes to warp') and the weft (*levado a tecer*, lit. 'that it takes to weave'). For the second, however, the legislator set the weight of the raw material for the weft without clarifying the composition of the warp. A question arises—were these mixed linen and woollen fabrics? Unfortunately, the answer is unknown. However, it is known that Portugal produced an abundance of linen, of the Galician and Moorish varieties, and that the fine threads and yarns spun in the kingdom supplied both regional and foreign markets [25] (p. 254). When cotton production was mechanised in the second half of the eighteenth century and problems were encountered with the resistance of cotton warps to the pressure of the sley, linen warps were used to overcome the obstacle [26] (pp. 150–151).

After weaving, the fabrics were refined with various finishing touches, namely fulling, dyeing, and shearing. Felting took place using a fullery, usually hydraulically powered [27] (pp. 5–6). These mechanisms combined compression of the material with water, fuller's earth, and/or other clays. This stage reduced the length of the fabric, giving the wool consistency, robustness, and impermeability [28] (pp. 107–108). Felting is most efficient when undertaken with thick fabrics made of loosely twisted yarns, as the openness of the fibre's scales facilitates the procedure. Of all the natural fibres, wool is the only one with this ability [29] (p. 443). After being dyed (Table 3), the fabrics were sheared. The aim of this stage was to standardise the texture of the fabric. Scissors greased with pork fat or chicken fat (*enxundia*, lit. kidney fat or suet) were used[17].

### 4.1. Pé de Azul

Dyeing is the second-most standardised process in the regulation (Table 3). The document stipulates that no *morados*[18], *leonados* (tawnies)[19], greens, or blacks could be dyed without a blue base [15] (pp. 32–34). The 'celestials', which resulted from the number of dips in the dye bath, classified the density of the indigo (using woad, indigo [obtained from plants] or both) and served as a standard for the black recipe (see Table 5). However, the regulation under study is silent on the method of obtaining these 'celestials'. Furthermore, the civilisations of Europe, India, and Persia codified the derivations of indigo according to the intensity of the blue. This particular standardisation system correlated a word to the saturation of the colour attributed by the number of immersions in the dye bath[20]. An identical method, also organised by celestials, operated during the Middle Ages in the Iberian Peninsula [34] and the Italian Peninsula [33] (p. 373). Between 1333 and 1480, the Leiden city guilds stipulated that black fabrics had to have been previously dyed with woad blue. The same determination was common to the most important medieval textile production centres such as the Flemish cities of Antwerp [35], Bruges, Ghent, and Mechelen [36] (pp. 67 and 76).

**Table 5.** Blue standards by type of cloth and colour, 1573/1690.

| Blue Standard | Type of Cloth and Colour |
|---|---|
| 5 celestials | Black fabrics with red selvedge |
| 3 celestials | Black fabrics with black selvedge |
| 1 celestial | *Baetas* ('baizes') dyed black |
| Turquoise [1] | *Dozenos* cloths dyed black |
| 1 ½ celestial | White cloths dyed black |

[1] The document reads 'toquejado'[21]. Source: [15].

Chapter LXXIV of the fabric regulation laid down a set of rules for 'blue dyers' [15] (pp. 35–36). However, there is no other known data that would confirm the existence of Portuguese dyers specialising in this colour, as had been the case in France since medieval times[22]. Although no blue fabric was required in the examination to obtain the trade of dyer in the city of Lisbon (1572)[23], a system of *pé de azul* (lit., 'blue foot', in the sense of 'blue base') was already in operation[24]. At the beginning of the eighteenth century, Rafael Bluteau pointed out that the 'domestic' woad (likely a subspecies of woad particular to Portugal—see p. 10 below) served 'as the foundation of all dyes' because, with it, the dyers prepared 'the fabrics to receive all the other colours' [22] (p. 310). The implementation of the blue base system was reflected in the architecture of European dyehouses and Portuguese manufactories. In 1788, the Royal Factory of Covilhã had specific vats for woad [38] (p. 99) with heating conditions ideal for the fermentation of indigo [39] (p. 95).

The functional separation of blue from the other colours is due to the specificities of the dyeing process. It is possible to colour textile raw materials using three methods based on different chemical assumptions. 'Direct dyeing' results from the preparation of

a bath in which the dyeing substance is mixed or boiled with the yarns or fabrics. This method is applied with rare dyes, such as those from lichens or green walnut shells, whose molecules orient themselves favourably toward those of the textile fibres. This type of dyeing is, however, more vulnerable to washing. A second system, 'vat dyeing', occurs when the colouring substance is not soluble in water. As such, an alkaline reduction bath must first be prepared to make the molecules soluble; this bath takes on a yellowish or greenish colour. When the textile fibres emerge and come into contact with oxygen, the colours precipitate and blue becomes visible. It is used for dyeing blue (present in woad and indigo leaves) or purple with molluscs. The third method results from the addition of metallic salts and is called 'mordant dyeing'. Most colouring substances do not bind strongly to textile fibres, but they combine well with metal salts. Certain substances, such as alum, cream of tartar, iron and copper sulphates, or plant species rich in tannins (sumac or gallnut, for example), act as intermediaries between the dyes and the fibres, binding them. It was customary for the mordant bath to precede the dye bath [33] (pp. 4–6).

The Egyptian civilisation (from the third century to the seventh century BCE) pioneered the dyeing of blue using woad [40] (pp. 29–30) whilst the Neolithic populations of western Europe also knew how to dye various shades of this colour. Archaeological finds from the Bronze Age, Iron Age, and Roman periods confirm the use of woad in textiles and ceramics [33] (p. 374). Written sources from antiquity document these uses, although the Greeks and Romans valued indigo above all [41] (p. 201). The woad industry became very popular in Europe from the thirteenth century onward, being at the confluence of advances in dyeing and changes in both patterns of production and consumption. With the extensive use of woad, it became possible to dye dense, saturated blues—the social elite, from clothes to paintings, assimilated these colours as a sumptuous and luxurious element [4] (pp. 53–69).

The granting of a privilege for the exploitation of woad in mainland Portugal, awarded in 1445 to Prince Henry the Navigator (1394–1460), mentions the novelty of this culture in the kingdom. In the overseas territories, woad production in the Azores was of strategic interest to international trade, as it was exported to Seville and London at the end of the fifteenth century [28] (pp. 116–117). Valentim Fernandes, at the beginning of the sixteenth century, recorded the production of the plant on Terceira Island[25]: it was sown in February; taking advantage of the reduced rainfall, the first leaves were then harvested in May until September. One documentary source mentions some details about its processing: once harvested, the leaves were taken to a mill; after some of the water had been released (the 'bad juice') and the pigment concentrated ('the leaf is left with its own virtue'), they were moulded into round loaves and set out to dry; once dry, they were ground into powder; after this grinding, they were *granados* (lit., 'granulated') in a house where '[they] pour water and stir it'. After these processes, the woad was ready to sell [42] (p. 152).

The commercial importance of this raw material and the need to ensure the quality of the crop led King João III to issue regulations in 1536. The monarch was 'informed that the woad that was harvested on the islands was too defective and not as perfect as it should be', thus running the risk of jeopardising the trade, which 'was large and could become much larger' [10] (p. 393). The monarch established supervisory figures, the *lealdadores*, appointed by the judges and officials of the municipalities. This document regulated the harvesting and processing of the woad in the phases already summarised by Valentim Fernandes in some detail. The woad leaves had to be harvested three or four times a year from mid-May to October, depending on the orographic characteristics of the production centres within the archipelago[26]. The raw material had to be picked half an hour after sunrise to avoid the humidity of the dew—it was believed that the water removed the colour from the plant[27]. That night or the day after the harvest, it was ground into a dough that was spread out on trays to prevent fermentation. Up to the third day, the dough was moulded into cakes (*embolar*), which were then laid out to dry in the shade on reeds. The last stage was the *granagem*—granulation—which was carried out with a millstone and clean water [10] (p. 398). The document does not provide any further details about this

stage of processing. Rafael Bluteau, in the first quarter of the eighteenth century, explains that once the woad was dry and the balls had been reduced to powder, 'it is soaked in water for four months and, during this time, it is stirred and turned about forty times' [22] (p. 311).

The processing seems similar to that used in the regions of Thuringia (Germany) or Languedoc (France)—Figures 1 and 2. In France, the *agranat* (dried woad) was fermented in buildings called *greniers*; the similarity with the word mentioned by Valentim Fernandes (*granar*) is obvious. In Thuringia, there were professionals who specialised in this very secret stage. After weeks of fermenting in these conditions, the woad would be formed into 'grains'. This raw material could be sold and henceforth be used for dyeing [33] (369–370).

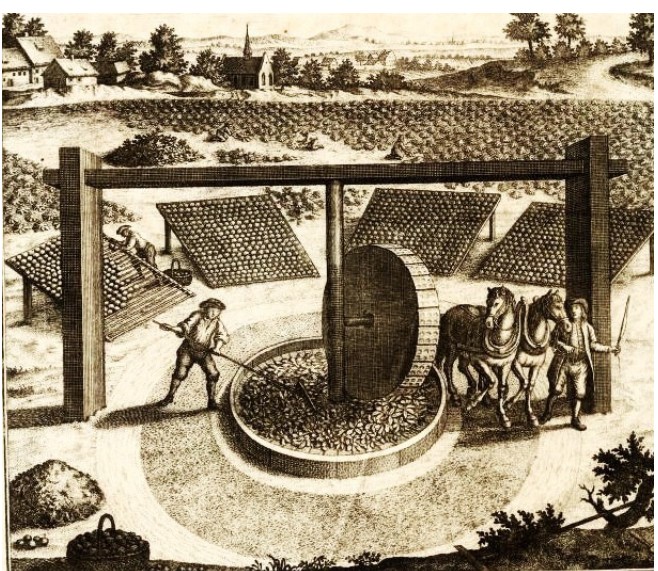

**Figure 1.** Illustration of a woad mill in Thuringia (Germany), 1752. Source: [43].

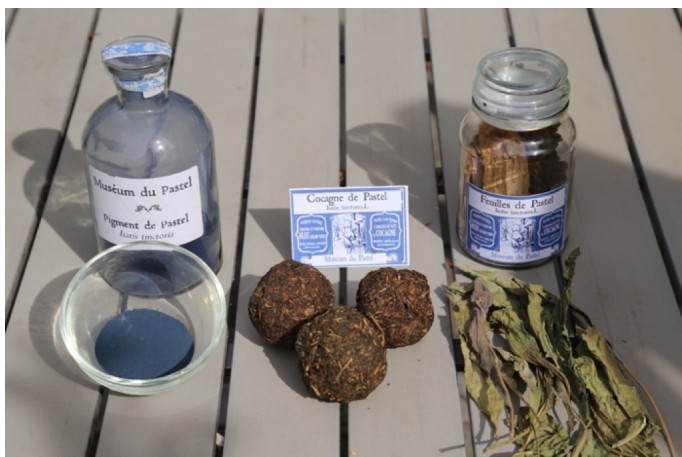

**Figure 2.** Dyers' woad according to the Woad Museum (Labège, France): powdered blue colouring, balls with crushed woad, and dried leaves of the *Isatis tinctoria* plant. Source: photo by Frédéric Neupont under licence CC BY-SA 4.0 DEED.

Duarte Nunes do Leão, in his *Descrição do Reino de Portugal* ('Description of the Kingdom of Portugal') (1610), mentioned that the Count of Portalegre, João da Silva, had successfully sown woad in a *lezíria*[28] in his territories in previous years [11] (p. 59v). In the sixteenth and seventeenth centuries, production from the Azores was both considerable and connected with large wool centres in England and the Netherlands, exchanging Portuguese woad for textiles from those regions [10]. In the mid-eighteenth century, woad from the

islands was traded at 1200 *réis* per *quintal* (ca. 60kg), around 20% of the average price of indigo from the Portuguese empire in the East[29]. This difference in value provides a pointer as to the importance of the product, which was used to dye medium- or low-quality cloths. The Portuguese production of woad was very significant. Miguel Gerónimo Suarez y Nuñez's *Arte de teñir las lanas, sedas, hilo y algodón...* (1779) suggests that, although woad was widespread in many European countries, there were three species of the plant: one from the French Lanquedoque; a wild one, identical to the first; and a 'smaller one, [that] is only encountered in Portugal' [44] (p. 243).

As shown, blue could be created with woad, indigo, or a mixture of both. Indigo is a plant that originated in India and was widespread in the Roman world [41] (p. 201). In the thirteenth century, documentary references to indigo appear in the Portuguese locations of Atouguia and Estremoz, as a tradable commodity that probably came from North Africa. From the fifteenth century onward, indigo was produced in Cape Verde and on the coast of Guinea. With the opening of the Atlantic connection to India, the Portuguese had direct access to this commodity [28] (p. 120). During the second half of the seventeenth century, the first attempts were made to introduce indigo factories in Brazil [45]. Rafael Bluteau, quoting Friar João dos Santos' *Ethiopia Oriental* (1609), described the processing of indigo from the Quirimbas islands, located 'sixty leagues' from Mozambique: '. . .a few days after they have harvested (...) [the] grass, they tread it very well and soak it in some troughs of water, where it is curing and fermenting, and there they stir it so that it crumbles. Once it is well broken up, it is boiled, then stirred and dissolved until it is like a paste and, after that, they throw it back into troughs or stone basins and put it to bask in the sun, where it curdles, and they take it out of there, in pieces, dry and hard as stone' [37] (p. 367).

As with granulated woad, the processed raw material could be exported and used in dye factories. In 1752, indigo from Castile and the Portuguese Empire of the East was traded in Portugal, moulded into the shape of figs or *telhas* (a kind of tile) (see Table 6). To dye it, a bath had to be prepared in a vat in order to create a water-soluble dye from the pigment that could then be incorporated into the textile fibres. The chemical reaction transforms an insoluble molecule (indigotine) into a soluble molecule (indoxyl) in water. Regardless of the specific solutions, the reduction and extraction of the dye takes place through the fermentation of the raw material in an alkaline environment. Certain agents, such as lime, ash, or sodium carbonate, neutralise the acids produced via fermentation and prevent oxidation. Once immersed in this bath, the textile fibres turn blue when exposed to oxygen. According to the regulations, woad dyers were not authorised to wash their cloths in salt water or to mix brazilwood[30] or lime in their recipes [15] (pp. 35–36).

**Table 6.** Market prices of raw materials used in textile dyeing, 1752.

| Common Name | Scientific Name | Price (*Réis*) | Unit | Price per Kilo (*Réis*) (a) |
|---|---|---|---|---|
| *Açafrão de Castela* (Castilian safflower) | *Carthamus tinctorius* | 2000 | *arrátel* | 4357.30 |
| *Açafrão de França* (French safflower) | *Carthamus tinctorius* | 2000 | *arrátel* | 4357.30 |
| *Alúmen (sulfato de alumínio e potássio)* (Alum—sulphate of aluminium and potassium) | - | 3000 | *quintal* | 50.74 |
| *Anil de Castela* (Castilian indigo) | *Indigofera tinctoria* L. | 65,000 | *quintal* | 1099.46 |
| *Anil de figos das Índias de Portugal* (Indigo in the shape of figs, from Portuguese India) | *Indigofera tinctoria* L. | 30,000 | *quintal* | 507.44 |
| *Anil de telhas das Índias de Portugal* (Indigo in the shape of tiles, from Portuguese India) | *Indigofera tinctoria* L. | 18,000 | *quintal* | 304.47 |
| *Caparrosa* (Sulphates) | - | 600 | *arroba* | 40.60 |

**Table 6.** *Cont.*

| Common Name | Scientific Name | Price (*Réis*) | Unit | Price per Kilo (*Réis*) (a) |
|---|---|---|---|---|
| *Cochonilha da terra em pasta* (Cochineal in paste, produced in Portugal) [1] | *Dactylopius coccus* | 800 | *arrátel* | 1742.92 |
| *Cochonilha de silvestre* (Wild cochineal) | *Dactylopius coccus* | 1500 | *arrátel* | 3267.97 |
| *Dragoeiro (sangue de dragão)* (Dragon blood tree) | *Dracaena cinnabari* | 80 | *arrátel* | 174.29 |
| *Fustete (ameixieira de espinho)* (Madeira barberry) | *Berberis maderensis* | 2000 | *quintal* | 33.83 |
| *Grã em folha de Olivença* (Kermes in sheets from Olivença) | *Kermes vermilio* | 6000 | *arroba* | 405.95 |
| *Grã em folha de Setúbal* (Kermes in sheets from Setúbal) | *Kermes vermilio* | 9000 | *arroba* | 608.93 |
| *Grã em pó de Olivença* (Powdered kermes from Olivença) | *Kermes vermilio* | 12,000 | *arroba* | 811.91 |
| *Grã em pó de Setúbal* (Powdered kermes from Setúbal) | *Kermes vermilio* | 18,000 | *arroba* | 1217.86 |
| *Grã* or *cochonilha da Índia* (Kermes or cochineal from India) | *Kermes vermilio* or *Dactylopius coccus* | 3600 | *arrátel* | 7843.14 |
| *Noz de galha* (Gallnuts) | *Quercus infectoria* | 8000 | *quintal* | 135.32 |
| *Pastel que vem das ilhas* (Woad from the islands, i.e., Azores) | *Isatis tinctoria* L. | 1200 | *quintal* | 20.30 |
| *Pau amarelo* (Dyer's mulberry) | *Morus tinctoria* | 400 | *quintal* | 6.77 |
| *Pau brasil* (Brazilwood) | *Caesalpinia sappan* L. | 50,000 | *quintal* | 845.74 |
| *Pau Campeche* (Logwood) | *Hematoxylon campechianum* | 4000 | *quintal* | 67.66 |
| *Ruiva* (Dyer's madder) | *Rubia tinctorum* L. | 4000 | *quintal* | 67.66 |
| *Sumagre* (Tanner's sumac) | *Rhus coriaria* L. | 480 | *arroba* | 32.48 |
| *Urzela que vem da Madeira* (Orchil from Madeira) | *Rocella tinctoria* | 600 | *arroba* | 40,60 |

[1] The expression 'da terra' suggests production on the Portuguese mainland. However, there is no other reference to cochineal breeding in Portugal. In his discussion on cochineals, Rafael Bluteau [66] points to production in the 'Indies of Castile', namely those arriving in the port of Cádiz (Spain) from Peru (called *mesteque*) and Mexico. It is possible that the use of the expression 'da terra' associated with cochineal results from a confusion with the *grã* (kermes) harvested in Portugal, as both species produce a similar red colouring that allowed the dyeing of vermilion. The author of the source equates them when he lists 'Grã or cochonilha da Índia' under the same price (3600 *réis* per *arrátel*). Source: [12] (pp. 166–204). Caption: (a) Given the variety of units of weight used, to allow comparisons between the different substances, a price per kilo (in *réis*) was reckoned with the help of Table A1.

### 4.2. Black Itself

Once the *vedor* had checked that the blue was 'as high and perfect' (*tão subido e perfeito*) as it should be, the cloths were ready to receive other colours. The regulations present four formulas for dyeing black cloths (Table 7) intended for the highest quality fabrics (*vintequatrenos* and *belartes*). The recipe could be applied, by adaption, to *vintedozenos*, *vintenos*, *dezochenos*, *guardeletes*, and *estamenhas*—see Table 4. The colour was therefore reserved for textiles with more than 1600 yarns in warp [15] (p. 27) and with a standard length of eight branches.

**Table 7.** Recipes for black cloths in the regulation, 1573/1690.

| Recipe | Woad or Indigo (Number of Celestials) | Alum (*Arrátel*) | Tartars (*Arrátel*) | Iron sulphate (*Arrátel*) | Sumac (*Arrátel*) | Castilian Madder (*Arrátel*) (c) | Flemish Madder (*Arrátel*) (c) |
|---|---|---|---|---|---|---|---|
| Black cloth with black selvedges (a) | 3 | 2 | 4 | 3 1/2 | 5 ou 6 | 50 | 40 |
| Black cloth with red selvedges I (b) | 5 | 4 | 5 | 3/4 | 3 | 50 | 40 |
| Black cloth with red selvedges II (b) | 7 | 5 | 4 | 1/2 | 2 1/2 | 75 | 70 |
| Black cloth with red selvedges III (b) | 9 | 5 | 3 | - | 1/2 | 100 | 84 |

Source: [15] (pp. 26–36). Captions: (a) Recipe for any cloth finer than a *dezocheno*; (b) Recipe for *vintequatreno* and *balarte* fabrics applied, by analogy, to *vintedozenos*, *vintenos*, *dezochenos*, *guardeletes*, and *estamenhas* as long as they have the same length (8 *ramos*) and respect the number of blues (Table 5); (c) Either could be used. Flemish madder had a greater concentration of dyeing agent. Note: black *baetas* (baizes) should follow the same recipe with the least amount of *boa palmilha* (lit., 'good quality *palmilha*') blue and 27 *arráteis* of madder.

Metallic salts (potassium aluminium sulphate [alum], *rasuras* [tartars][31], and iron sulphate) and tannin-rich plants (sumac) acted as mordants and tone-darkeners. The ratio between alum and iron sulphate (*caparrosa*), the quantity of tannins (sumac), and the weight of red dyes (madder) varied according to the amount of blue (i.e., number of celestials) and the desired intensity and saturation of the black. As Table 7 makes clear, red dye and alum followed the blue standards upward whilst cream of tartar, plants with tannins, and iron sulphate had the opposite trend. It is believed that the recipes would produce shades of black with hints of violet or purple, as the designations of the cloths 'black with red selvedges' and 'black with black selvedges' suggest. The name 'black' or related words could include different nuances depending on the colouring material used in the recipe. European documents from the fifteenth and sixteenth centuries use words for colours corresponding to reddish blacks, brownish blacks, or greenish blacks [35]. Combining the red of madder with indigo was a solution often used by European dyers to achieve shades that imitated purple from molluscs, scarlet from insects, or to dye blacks [33] (pp. 114–115).

Back to the recipe: how to make it? Firstly, corresponding to the mordanting itself, the cloth was immersed in the *umar* (the dyeing mixture) made up of alum[32], tartar, and iron sulphate, which was boiled for four hours. The cloth had to be stirred without rest. The proportion of these substances could vary according to the 'quality' (pH) of the water. The cloth was then removed and left to dry, covered and sheltered, on a structure known as a *cavalo* ('horse'). The next day, the cauldron was filled with clean, cold water to which the sumac was added[33]. This was then stirred for half an hour. Next, over a low heat, the madder was added. As soon as it began to boil, the cloth was dunked into the bath where it was left to rest for 15 min [15] (pp. 27–28). The cloth would thus be dyed black.

The proportion for the mordanting was 15% alum with 6% cream of tartar (21% of the total) or 15–25% alum in relation to the weight of the dry wool [33] (pp. 12–13). Table 8 summarises the calculations undertaken in this study to correlate weights of the mordant substances against woven wool. As the weight of the warps of some typologies is unknown, a comparison was made with cloths where this variable is available (see Table 9). Given the changeability between the four recipes according to the blue base and the weighting of fabrics with different masses, average percentages were calculated for the finest fabrics, the *vintequatrenos dezimados*. It was found that the averages for alum (6%) were lower than those estimated by Dominique Cardon but closer when added to the average values for sumac (4%) and iron sulphate (2%). The average for tartar (6%) is very close to the standard. Overall, the weight of the mordants and ancillaries in relation to the weight of the fabric (18%) is within the margins stated by the literature. Sumac was widely used in leather

tanning and had mordant and colour-enhancing properties [33] (pp. 431–434). After the *umar* (the mordent mixture) and the *cascarrear* (the subsequent sumac bath), the black colour was achieved with the addition of madder. The weight of this dye in relation to the weight of the wool was high: 95% in the case of Castile madder and 81% when it was Flemish. The roots of this plant made it possible to produce various shades of red.

**Table 8.** Ratios of dyestuffs (%) in the weight of wool (72 *arráteis*) in *vintequatrenos dezimados* cloths, 1573/1690.

| | Base of Wood or Indigo (number of Celestials) | Alum (*Arrátel*) | Tartars (Cream of Tartar) (*Arrátel*) | Iron Sulphate (*Arrátel*) | Sumac (*Arrátel*) | Castilian Madder (*Rubia tinctorum* L.) (*Arrátel*) | Flemish Madder (*Rubia tinctorum* L.) (*Arrátel*) |
|---|---|---|---|---|---|---|---|
| Black cloth with black selvedges | 3 | 3% | 6% | 5% | 8% (a) | 69% | 56% |
| Black cloth with red selvedges | 5 | 6% | 7% | 1% | 4% | 69% | 56% |
| Black cloth with red selvedges | 7 | 7% | 6% | 1% | 3% | 104% | 97% |
| Black cloth with red selvedges | 9 | 7% | 4% | 0% | 1% | 139% | 117% |
| Average | - | 6% | 6% | 2% | 4% | 95% | 81% |

Source: [15] (pp. 26–36). Caption: (a) The recipe indicates five or six *arráteis*; here, an average value of 5.5 was used.

**Table 9.** Weight of the wools in cloths called *dezimados*, 1573/1690.

| Designation of the Cloth | Yarns in Warp (Number) | Weight per Ramo (*Arrátel*) | Weight of an Eight-*Ramos* Piece of Cloth (*Arrátel*) |
|---|---|---|---|
| *Dezocheno dezimado* | 1800 | 7 | 56 |
| *Vinteno dezimado* | 2000 | 7 ¾ | 62 |
| *Vintedozeno dezimado* | 2200 | 8 ¼ | 66 |
| *Vintequatreno dezimado* | 2400 | 9 | 72 |

Source: [15] (pp. 26–36).

The presence of madder (or *garança*) in Portuguese territory can be traced to truly ancient times. The oldest textile fragment found on the Iberian Peninsula was discovered in the megalithic necropolis of *Belle France* (Caldas de Monchique). It corresponds to a fine linen fabric painted with madder, dating from the middle of the third millennium BCE [48]. Classical sources indicate that crushed madder roots were used to obtain shades of red. It was widely used by the Germanic peoples who created the colour through a vegetable-based dye, differing from the animal-based red (purpura lapillus and the murex [sea snail]) popular in Roman civilisation [41] (p. 202). In the Middle Ages, the madder that reached the kingdom of Portugal was imported from Castile [28] (p. 115). The 1690 regulations mention the existence of a species with a higher concentration of colouring matter originating in Flanders—smaller quantities were needed compared to the norm to achieve the same result in terms of dyeing (see Table 7), this being explained by the existence of different species of the same plant with varying concentrations of red and yellow dyes[34]. By the mid-eighteenth century, madder was commercialised at 4000 *réis* per *quintal* (see Table 6); it is not known if this was the one that was grown in Coimbra, Castelo Branco, Idanha, and Covilhã, and that by the second half of the century reached the *Real Fábrica da Covilhã* [39] (p. 95). However, the national production of madder was minimal by the 1840s[35]. The roots

of this plant made it possible to dye various reds, imitate vermilion, and achieve shades of pink. When combined with blue, purple or black tones could be created and, when mixed with tannins, shades of brown could be obtained. Orange colours could be achieved when mixed with flavonoid-rich plants such as the dyer's lily. This versatility of madder was taken advantage of by the largest dyeing centres in Europe [33] (p. 113–116).

In Portugal, another source of red was imported and produced in the Early Modern Age: the *grã* (*Kermes Vermilio*). This 'true bug', a scale insect of the order *Hemiptera*, is a parasite of various plants of the *Quercus* genus and was a luxurious and highly valued colouring agent because it dyed scarlets—commonly known as vermillion. The *Pragmática* [lit. praxis or customs] law of 1340 stipulated that scarlet cloths were the privilege of the Portuguese royal family [50]. In the fifteenth century, insects were harvested in Alcácer do Sal, Sintra, Sesimbra, Setúbal, and Olivença [28] (pp. 113–114). The luxurious status of insect-dyed fabrics still finds support in the regulations because only the best quality cloths (2400 threads) could be dyed with this dye [15] (pp. 26–27). Duarte Nunes do Leão, at the beginning of the seventeenth century, mentions that Pliny the Elder (23 CE–79 CE) praised the Lusitanian *grã* as the finest and best of its time. Duarte Nunes do Leão confuses it with a vegetable colouring, perhaps because it is a parasite of the *quercus coccifera* (known precisely as 'kermes oak') but writes that the *grã* referred to by Pliny is the same that was still being harvested in the seventeenth century in the Arrábida and São Luís mountains, in the region of Setúbal. He also mentions a production located in Aljustrel [11] (p. 59v). Rafael Bluteau refers to the Sesimbra *Kermes Vermilio* as being the best in Europe [51] (p. 106). In 1752, *grã* from Olivença, Setúbal, and the Indies were traded, processed into a powder, leaves, or paste. In the mid-eighteenth century, these scale insects, along with safflower (*Carthamus tinctorius*), indigo, and brazilwood, were the colouring agents commanding the highest commercial value (see Table 6).

Medieval European dyers created a type of black of great prestige—called *brunette*, *brunetta* or brunet—through a blue base followed by a bath of mordant with alum, tartar, and madder [46] (p. 475). The Valencian Joanot Valero's dyeing manual (fifteenth century) mentions a recipe for *negro à la contraya* (Courtray-style black)[36] that is very similar to that found in the studied regulations [6] (pp. 235–236). The prescriptions for dyeing black in western Europe between 1650 and 1850 were divided into three large groupings. The first dictated a vat dyeing with woad or indigo followed by another dyeing using mordants (alum and/or tartar) and a red dye from madder. Yellow dyes could be added to save money or to change the tone of the black. The second group used tannins (gallnut, sumac, alder, or redoul [*Coriaria myrtifolia*] leaves) combined with metallic salts such as green or red sulphates (iron or copper); however, excessive use of iron sulphate caused damage to the wool fibre. The third group of formulas involved combining one or more sources of indigo (indigo, woad, or logwood) with tannins (sumac, gallnut, alder, or walnut bark) and metallic salts to which red (madder or brazilwood) or yellow (dyer's lily or broom) dyes were added to darken and tone the black [52] (pp. 118–119).

Why black? The attention paid by the regulations toward dyeing this colour is not without reason—to summarise, as this issue will be addressed in another text, the answer to this question will be brief and merely exploratory.

The explanation likely lies between the chronology of the domestication of dark colours by European dyeing, making it possible to achieve dark and saturated tones, and the social and economic capital acquired by the colour between the end of the fourteenth century and the dawn of the Early Modern period. Black is a colour unlike any other. Not only because Isaac Newton (1643–1727) stripped it of its status as a colour [53] (p. 200–206), but furthermore, it cannot be obtained as a dye without the involvement of several raw materials. In the mid-nineteenth century, João Baptista Lúcio explained that 'all the black colours that the dyer obtains are due to an artificial combination: the black molecules that he forms by combining the astringent principle or another colouring substance with iron oxide are fixed on the wool' [54] (p. 325). As has been shown, formulas for achieving dense,

dark tones revealed an advanced level of chemical knowledge, as they combined various types of colouring methods.

The combination of these technological factors with changes in taste and mentality seems to have conditioned the patterns of consumption of dyed cloths produced in the European luxury markets of the late Middle Ages [55]. The production of woollen cloth in Flanders, for example, shows a change: from the 1430s onward, dark colours (blues, greens, greys, and purples) began to outshine the bright, saturated colours (vermilion, red, white, and green) *en vogue* in previous centuries [36] (pp. 55–76). In Valencia, between 1475 and 1513, 73.47% of the silk fabrics sold in that market were black [56] (p. 88). John H. Munro [36] (pp. 87–88) equates the decisive role of Portugal and Castile in the circulation of this taste. It seems credible: the Duke of Burgundy, Philip the Good (r.1419–1467), to whom the fashion for black is commonly attributed, married Isabel of Portugal, the daughter of King João I. King Duarte, Isabel's brother, wore black to mourn their father's death in 1433 [57] and his son, King Afonso V, was depicted, aged 25, dressed in black (Figure 3). Between December 1464 and August of the following year, the Pisan company Salviati-Da Colle sold 75 silks in the Lisbon market; 51% of the fabrics traded were black, and one of the clients was King Afonso V [58] (p. 53). The economic interest in these dark colours must have grown when, during the sixteenth century, society increased its interest in the discretion, sobriety, and simplicity of black, in keeping with the precepts of the Reformation [59] and Counter-Reformation. In the case of the reformed Catholic Church, the assumption between moral virtue and black was internalised in the appearance of priests, which was regulated by bishops through synods. The wearing of black robes by secular Catholic clerics signalled their virtuousness and intellectual superiority [60]. Interest in this colour cut a swathe across the various strata in Early Modern Europe [61].

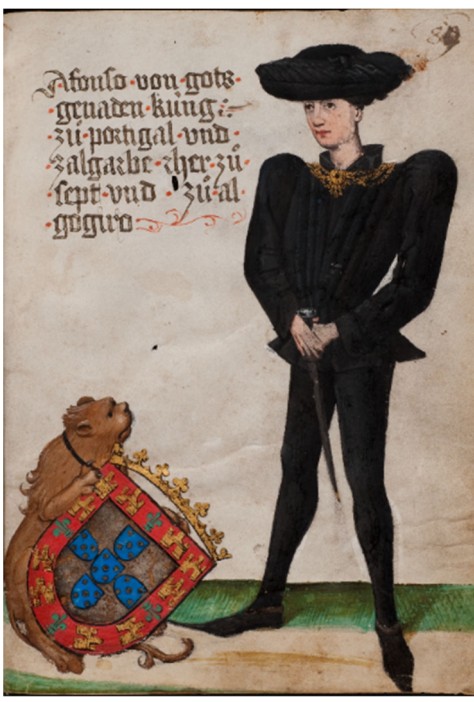

**Figure 3.** King Afonso V of Portugal (1432–r.1446–1481). Source: *Württembergische Landesbibliothek, Ritter Georgs von Ehingen Selbstbiographie, 1455*, Cod.hist.qt.141, fl. 89. Public Domain Mark 1.0.

The fact that the Portuguese *Regimento* firmly established that black was to be used with the best quality cloths seems to corroborate the sumptuary nature of fine fabrics dyed in dark colours. Dark and lustrous blacks combined the apparently antagonistic variables of luxury—as a demarcation of social hierarchy—with moral edification.

## 5. Conclusions

Black was domesticated in two stages. The first of these processes was a preliminary bath in blue. A number of *celestes* (celestials) corresponded to the saturation and density of the indigo achieved by the various immersions of the cloth in the vat. In order to standardise this variable and guarantee the quality of the black, the king distributed standards to the wool-producing municipalities. These models were to be reviewed every three years with the participation of officials from Covilhã, Portalegre, and Estremoz. As blue was exogenous to the regulations, it was not possible herein to demonstrate the procedures used to assemble this 'indigo vat'. Nevertheless, it was possible to determine that Portugal had an important production centre for woad cakes in the Azores, and that it also received indigo from other parts of its empire. It is not entirely clear how the indigo was fermented, as the use of lime was forbidden, and moreover, the quantities of the raw colouring material used or the care taken to correctly maintain the stability of the indigo vat are both unknown. The system that prescribed blue as the foundation of all colours had implications for the architectures of factories and workshops, the organisation of work, the specialisation of the dyeing profession, and the movement of raw materials in regional and international markets.

Once the cloths had been dyed blue, they were moved to a second stage, which the regulations detailed in four recipes with particular care. The cloth would be 'wetted' with alum, tartar, and iron sulphate, and then 'coated' with sumac, which, as a tannin, would perform the dual function of mordant and colour enhancer. Then, the dyers' madder was added. A documentary analysis showed that an increase in the shade of blue led to an increase in red colouring and alum. In opposition, the quantities of tartar, sumac, and iron sulphate would decrease.

The recipe for dyeing black, as set out in the regulations, is not, in its general principles, a characteristic Portuguese trait. This formulaic solution has been shared by the main dyeing centres in Western Europe since the fifteenth century and throughout the Early Modern Period. The recipes presented varied according to the density of the blue and in the tonality and saturation of the black desired. The research undertaken suggests that three of the formulas would result in a black with violet or purple hues, with the other producing a denser and more saturated black. The system of naming and socially categorising dark colours by their hues was common in Europe in the Early Modern Age and was justified by the raw materials involved in dyeing. In this author's opinion, the cultural shift from the sumptuousness of vibrant and open colours, such as scarlet reds, greens, and whites, to densely saturated, dark tones justifies the centrality of black in the regulation. The religious reforms of the sixteenth century intensified the moral tilt toward anti-chromatic tones. The social and cultural values of the sixteenth century accentuated the stigma toward reds, greens, and yellows and deepened the interest in dark colours, considered to edify and systematise moral virtues. The establishment of rules for the dyeing of fine fabrics with dark colours and the use of recipes associated, in other European recipe books, with Flemish black suggest an intention by the Portuguese crown to imitate the high-quality manufactures of Western Europe and maximise the value of the wool produced in Portugal.

**Funding:** This text was written within the scope of the doctoral research 'Pobres, doentes e esfarrapados? Indumentária de pobres no contexto assistencial urbano de Porto e Lisbon (séculos XVII–XVIII)' funded by the Fundação para a Ciência e Tecnologia (ref. 2020.04746.BD). This work received support from PT national funds (FCT/MCTES, Fundação para a Ciência e Tecnologia and Ministério da Ciência, Tecnologia e Ensino Superior) through the project REVIVE (ref. 2022.01243; https://doi.org/10.54499/2022.01243.PTDC, accessed on 5 January 2023).

**Acknowledgments:** The artist Anafaia Supico asked me to write a short text on the history of natural textile dyes in Portugal. The aim was to use this work as a scientific dissemination in the context of an exhibition by the Siroco Collective at the Wool Museum in Covilhã (Portugal). Through various vicissitudes, that question turned into the pages of this article. It also contributed to my long-standing interest in the colour black in the context of the central area of my research (the history of clothing, costumes, and fashion). Wearing black has meant, at various times in history, the defence of the rational, disciplined, and ascetic body. Is it possible for human beings to control their animal nature and the transparency of their emotions through the use of a colour? The interpellation fuelled the fire of knowledge and conditioned my approach. Maria Marta Lobo de Araújo and Maria Augusta Lima Cruz made structural contributions to the discussion of the historical context of the regulations during the Early Modern Age. Joana Sequeira and Natalia Ortega Saez suggested sources and bibliography. Alice Bernardo, from the SaberFazer.org project, clarified my doubts, suggested resources, and provided bibliography on indigo dyeing. The artist Faia and the artisans Guida Fonseca and Kiri Miyazaki, who taught me how to dye, are the symbolic co-authors of these lines.

**Conflicts of Interest:** The author declares no conflicts of interest. The funders had no role in the design of the study; in the collection, analysis, or interpretation of data; in the writing of the manuscript; or in the decision to publish the results.

## Appendix A

**Table A1.** Correspondence for converting weight and length units used in the *Regimento dos panos*.

| Original Unit | Correspondence |
|---|---|
| *Almude* | 25 L (a) |
| *Arrátel* | 459 g (a) |
| *Arroba* | 14.780 kg (a) |
| *Canada* | 1/10 of almude (a) |
| *Côvado* | 0.66 m (a) |
| *Onça* | 28.691 g (a) |
| *Quintal* | 4 arrobas (a) |
| *Ramo* | 6 1/3 côvados (b) |

Sources: (a) [62]; (b) [15] (p. 7).

**Table A2.** Dyestuffs [1] used in the Portuguese territory during the Early Modern Age.

| Common Name | Scientific Name (b) | *Regimento dos Tintureiros* **(1572)** | *Regimento dos Panos* **(1573/1690)** | **Customs Tariff (1752)** |
|---|---|---|---|---|
| *Açafrão Bastardo* (Safflower) | *Carthamus tinctorius* | - | - | X |
| *Anil/Indicum* (Indigo) | *Indigofera tinctoria* L. | X | X | X |
| *Cochonilha* (Cochineal) | *Dactylopius coccus* | - | - | X |
| *Dragoeiro* (Dragon blood tree) | *Dracaena cinnabari* | - | - | X |
| *Fustete* (Madeira barberry) | *Berberis maderensis* | - | X | X |
| *Grã/Coccum* (Kermes) | *Kermes vermilio* *Kermes ilicis* L. | X | X | X |
| *Lírio dos Tintureiros/Lutum* (Dyer's rocket) | *Reseda luteola* L. | - | X | - |
| *Noz de galha/bugalhos* (a) (Gallnut) | *Quercus infectoria* | X | - | X |
| *Pastel dos tintureiros/Glastum* (Dyer's woad) | *Isatis tinctoria* L. | X | - | X |

**Table A2.** *Cont.*

| Common Name | Scientific Name (b) | *Regimento dos Tintureiros* (1572) | *Regimento dos Panos* (1573/1690) | Customs Tariff (1752) |
|---|---|---|---|---|
| *Pau amarelo* (Dyer's mulberry/Old fustic) | *Morus tinctoria* | - | - | X |
| *Pau brasil* (Brazilwood) | *Caesalpinia sappan* L. | X | X | X |
| *Pau campeche* (Logwood) | *Hematoxylon campechianum* | X | X | X |
| *Ruiva/grança* (Dyer's madder) | *Rubia tinctorum* L. | - | X | X |
| *Sumagre* (a) (Tanner's sumac) | *Rhus coriaria* L. | X | X | X |
| *Trovisco* (Flax-leaved daphne) | *Daphne gnidium* | X | X | - |
| *Urzela* (Orchil) | *Roccella tinctoria* | - | - | X |

[1] An effort was made to associate the common names used in the sources with scientific names in order to standardise the discourse in this text; it was therefore decided to name the most representative species within the same genus, although others may be included under the same name. This may be the case, for example, with indigo, woad, madder, scale insects, or tanner's sumac. Sources: Arquivo Municipal de Lisbon (AML), Casa dos Vinte e Quatro, *Livro de Regimentos dos Ofícios Mecânicos da cidade de Lisbon reformados por ordem do Senado por Duarte Nunes do Leão*, docs. 1–99, fls. 210v–231v; [15] (pp. 26–36); [12] (pp. 166–204). Captions: (a) Tannin-rich plants. They can also be used as mordants—complementary substances that can make hues darker; (b) The work by Dominique Cardon, *Natural Dyes: Sources, Tradition, Technology and Science* (Archetype, 2007), was used to identify the scientific names of stuffs.

**Table A3.** Mordant substances used in the Portuguese territory during the Early Modern Age.

| Common name | *Regimento dos Tintureiros* (1572) | *Regimento dos Panos* (1573/1690) | Customs Tariff (1752) |
|---|---|---|---|
| *Alúmen* (Alum) | X | X | X |
| *Caparrosa verde* (Iron sulphate) | - | X | X |
| *Rasuras* (Tartars) | - | X | - |

Sources: See Table A2.

# Notes

1. This author has recently found—and is currently studying—a five-volume work by João Baptista Lúcio, published in 1844.

2. As part of her doctoral thesis in History, Joana Sequeira presented several finds regarding the colouring raw materials in the Portuguese Middle Ages. Her study will be cited repeatedly throughout this article. Other works published in the first decade of this century should also be consulted on this matter [7–9].

3. The full title of the document is *Regimento sobre o beneficiar do pastel e eleição dos lealdadores*, dated 3 November 1536. This document was transcribed and published by Maria Olímpia da Rocha Gil in 1981 [10].

4. This author preferably consulted the manuscript version, although this source has been transcribed and published by Vergílio Correia in 1926. See notes 17 and 30.

5. By which the tax to be paid to the crown for the transit of products through the kingdom's dry and wet ports was established. Several dyeing substances were included in this list (*Pauta que há-de servir nas alfândegas destes reinos para o despacho dos portos secos, molhados e vedados que hoje corre por conta da fazenda real*, that is, 'Tariff to be used in the customs of these kingdoms for the dispatch at dry, wet and sealed harbours, which is currently under the royal treasury' 1830).

6. The spelling in the documents was updated to facilitate the understanding of the text and translation.

7. Giorgio Riello proposes a chronology for the production and consumption of this textile fibre. The first revolution in the production of cotton, characterised by a centrifugal system, began around the year 1000 and lasted until 1500 CE. An intermediate period, between 1500 and 1750, saw cotton become 'global', with India remaining the main centre for its processing and production. A second cotton revolution, starting around 1750 and lasting until the year 2000, is characterised by the separation, on a global scale, of the places of production and processing of the raw material; competition between American and Indian cotton plantations; the concentration of textile processing in the metropolises of the European empires; the democratisation of the consumption of cotton textiles; a change in consumption patterns; and the loss of market leadership by wool manufacturers [13] (pp. 187 ff.).

8      Mauveine, also known as aniline purple or Perkin's mauve, was the first organic synthetic dye. It was found in 1856 by William Henry Perkin and ushered in a new era in dyeing [14] (p. 22).

9      This is the purpose of the table that allows for the conversion of the information into current weights and measurements (see Table A1).

10     This document was published and commented on, in parts, in the bulletin *Lanifícios* published by the Federação Nacional dos Industriais de Lanifícios (National Federation of Wool Industries). The publication, edited by João Ubach Chaves, came out between 1950 and 1974. The same version was republished by Gil do Monte (pseudonym of Feliciano José Pássaro) in an appendix to his book *Fabricação de panos de cor e de linho em Évora e seu termo (século XIV a XIX)*, as both state that 'our bulletin is going to publish, little by little, in homeopathic doses, the famous *Regimento dos Trapeiros* of 1690' [5] (p. 1).

11     The high economic profitability of cotton, the possibility of growing it in the colonies at low cost and using slave labour, the industrialisation of spinning and weaving in the metropolises, and changes in consumption, with the growing appreciation of the colourful prints and the coolness and lightness of these textiles, explain why cotton fabrics supplanted woollen ones throughout the nineteenth century [13] (pp. 187–287).

12     This system was constituted by the *ponto* (1/12 of *linha*; 0.19 mm), *linha* (1/12 of *polegada*; 2.29 mm); *polegada* (2.75 cm), *palmo* (8 *polegadas*; 0.22 m), *côvado* (3 *palmos*; 0.66 m), and *vara* (5 *palmos*; 1.10 m). The *côvado* was used for dyed fabrics and the *vara* for undyed ones [21].

13     Open up the wool with an *escarduça* ('large card').

14     Once the fleece was laid out on a table, the wools were sorted into five categories: the 'edge' wool, i.e., that from the extremities, which had greater contact with dirt, and would be used for the selvedge of the cloth; then the first *sorte* ('kind'), cut three fingers long all round, was used for the fabric of *dozenos*; the second category of wool, cut three fingers higher, was used for *quatorzenos* and *sezenos* fabrics; the third wool, corresponding to the loin and the neck of the animal, was used for *dezochenos* and *vintenos*; and the fourth and noblest wool came from the flanks of the animal, and was used to weave *vintedozenos* and *vintequatrenos* fabrics [15] (p. 4).

15     The more varied the combs, the greater the range of cloths produced (see Table 4). The regulation established census criteria for accumulating combs. Weavers with assets of 200,000 *réis* or more could have up to five combs; up to 150,000 *réis* allowed them to accumulate four different combs; and 20,000 to 50,000 *réis* allowed them to have just one comb [15] (p. 12).

16     The chapters added in the seventeenth century have been excluded.

17     This step was done after dyeing [15] (pp. 36–37).

18     This is likely to be a blackberry shade between purple and black [30] (p. 317), although Rafael Bluteau claims it is the same as the colour *pardo* ('drab') [31] (p. 574). *Pardo* was a palette that included colours between beige, yellow, and brown. This author believes it parallels the natural colours of wool, although it could be dyed [32] (pp. 121–124).

19     'Colour similar to russet, like lion's hair' [31].

20     In India, the gradation of blue included seven levels. The oldest documentary evidence of this system corresponds to a cuneiform inscription dating from the seventh century BCE from the Second Babylonian Empire [33] (pp. 359–360).

21     According to Rafael Bluteau [37] (p. 698), there were three types of blue that varied in their saturation and density of colour: a very light one called *celeste* or *turqui*; another, darker and duller, called *azul-ferrete*; and a final one, called *azul ultramarino*, which painters used.

22     Medieval French masters were divided into blue dyers, who regularly dyed black and green, and red dyers. Their coexistence is marked by conflicts and disputes [4] (pp. 70–76).

23     Aspiring dyers were assessed on fine red cloth, fine yellow cloth, tawny low [quality] cloth, green *palmilha* (lit. 'insole', so probably a robust fabric made specifically for that purpose), black cloth, orange, tan, purple, and 'drab' pieces. Arquivo Municipal de Lisbob (hereafter AML), AML, Casa dos Vinte e Quatro, *Livro de Regimentos dos Ofícios Mecânicos da cidade de Lisbon reformados por ordem do Senado por Duarte Nunes do Leão*, docs. 1–99, fl. 211.

24     '[They] shall not dye any cloth in black from *quatorzeno* upwards without it first being blue'; '[they] shall not dye any clothing in wool or linen except with woad or indigo'; '[they] shall not dye any white cloth purple without it first being blue'. AML, Casa dos Vinte e Quatro, *Livro de Regimentos dos Ofícios Mecânicos da cidade de Lisbon reformados por ordem do Senado por Duarte Nunes do Leão*, docs. 1–99, fl. 212.

25     The author would like to thank Dr Joana Sequeira for her reference to this source.

26     The norm would be three harvests a year between May and September. An exception was made for some locations on the island of São Miguel, such as Rosto de Cão, Lagoa, and Pico de João Ramos, which, due to the higher temperature, allowed for four harvests. In the higher altitude areas, due to the low temperatures, woad grew more slowly but with better quality. In these areas, it could be harvested until 8 October [10] (pp. 394–395).

27     Concerns about the harmful effects of humidity extended to buildings used for grinding, moulding, and granulating. These structures had to be covered with straw and their walls mortared with lime or slabbed with stone [10] (pp. 397–398).

28   Banks of a fresh-water river that meets salt water: in the zone between the water types, flocculation between salinities occurs, leading to the precipitation of clays suspended in water and thus great fertility.

29   In total, 24,000 *réis* was the average price per *quintal* for Indies indigo in Portugal. See Table 6.

30   This addition probably served to intensify or darken the colour, resulting in a 'fake' blue. As a manuscript by a French dyer from the eighteenth century suggests: 'Blues are sometimes intensified or darkened with brazilwood and rosewood by putting a little alum in the bath to fix them, but this is a false dye that proves very damaging to the colour' [46] (p. 115).

31   *Rasuras* were the scrapings from the bottom of wine barrels: the tartar from a vessel where it must have fermented [34] (p. 70).

32   The use of this twofold sulphate of aluminium and potassium was common in European dyeing. At the end of the Middle Ages, alum was produced in Murcia and imported from regions in present-day Turkey, Italy, and North Africa [47] (p. 63).

33   In the dyers' regulation of 1572, this procedure was called *cascarrear*. See AML, Casa dos Vinte e Quatro, *Livro de Regimentos dos Ofícios Mecânicos da cidade de Lisbon reformados por ordem do Senado por Duarte Nunes do Leão*, docs. 1–99, fl. 212.

34   Baptista Lúcio (1844) mentioned the existence of three classes of madder: one with a higher concentration of red than yellow, grown in India, Cyprus, and Persia; an intermediate class, grown on the Italian peninsula and in France, with equivalent values between the two colours; and a third quality, produced in Spain, Alsace, Holland, Saxony, and Scotland, which had more yellow than red [49] (p. 60).

35   'As for the [madder] from Portugal, which has not, of late, been cultivated, we cannot comment on its quality' [49] (p. 60).

36   Courtray is a Flemish town, located in present-day Belgium.

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
