# Peer review of "Domesticating Colour in the Early Modern Age: Dyeing Wool in Black in Portugal"

_heritage, doi:10.3390/heritage7020042_

Round 1

Reviewer 1 Report

Comments and Suggestions for Authors

The paper is well written and explains clearly the aims and objectives stated at the introduction.

However, I find the use of notes within the text excessive and unnecessary. In many cases they are too long and do not contribute to the comprehension of the text as they refer to issues that are not clearly connected to your topic. I strongly suggest to review the notes and add those you find crucial in your main text and delete the rest of them.

For example, note 1 and 2 in the text the dates are given and the reader will not understand something more by reading these notes.

Author Response

Firstly, I would like to thank you for your careful observations. We have removed footnotes such as direct citations to sources that were only intended to prove a detail (footnotes 14, 17 or 37) or redundant information (footnotes 1 and 2). Other information, such as that in footnote 3, has been moved to the body of the text. This restructuring was carried out without reducing the clarity of the ideas or the cohesion of the arguments presented.

Changes resulting from this and other evaluations have been marked in yellow.

The author.

Reviewer 2 Report

Comments and Suggestions for Authors

The paper presents interesting research on the use of the colour black in textiles in Portugal.

I am a chemist, so my opinion cannot go into the assessment of the quality of the historical research.

However, I think it is a very well written and very interesting work. In my opinion it is rigorous and well documented.

The chemical aspects of the work are correct and interesting.

The use of tables is effective and makes the text very clear to follow.

The English language does not need revision.

In my opinion, the bibliography cited is complete and there is only one self-citation.

Below are some specific comments:

Lines 92-93

"(Tables 6 and 76 and approach their social value through the price of some of these raw materials (Table 8)"

I believe that a closed parenthesis is missing after number 7.

Lines 419-420

"Metallic salts (potassium aluminium sulphate [alum], rasuras [tartars])39 and iron sulphate) and tannin-rich plants (sumac) acted as mordants and tone-darkeners."

Please check the brackets in this sentence.

Line 663

"acetic"

I think it is 'ascetic'

Author Response

Thank you for your comments and suggestions. I feel satisfied that the text has no errors in the field of chemistry, which was one of the biggest challenges I felt when writing the work.
With regard to the indications on lines 92 and 93 and following comments from other reviewers, the citations to the tables in this part of the work have been removed. 
With regard to lines 419-420 and 663, the information has been corrected.
The changes resulting from this and the other reviews have been signalled in yellow.
The author.

Reviewer 3 Report

Comments and Suggestions for Authors

The following minor comments required to be reflected in the revision.

1.  Figure 1~3 : Did the authors get the permission of these pictures from the original copyright ?

2.  Table 6 : It will be better by showing the color shade for individual dyestuff in Table 6.

3.  Table 6~11 : The place of each table can be rearranged to the position that discussed and cited.

4. The conclusion part is recommended to be compact. 

Author Response

Thank you for your comments. The review process is very gratifying, precisely because it allows us to think about data and narratives in a more complex and complex way.

Regarding the images, they are all in the public domain under the licences referenced. However, I have clarified the ownership and licence of Figure 2 and, in the note for Figure 1, we have provided a link to the Internet Archive.

With regard to the shades/colours produced by the different dyestuff, it seemed to us at first that the suggestion was interesting. However, I found it difficult to simplify this information. Firstly, there are dyes that have the precursor of several colours. Secondly, all dyes can be mixed to produce different colours, which makes natural dyeing a virtually endless field of experimentation. Thirdly, iron salts or tannin-rich plants, for example, take on different functions depending on the dosage and combination with other raw materials.

In some cases, these agents not only tone and darken the tones, but also produce colours that are completely different from those that the dye would produce when used alone. For these reasons, I believe the issue is too complex to be generalised in a table.

We recognise that the original organisation of the tables was poorly constructed. We have therefore reorganised them according to the order in which they are cited in the text. We have placed three tables in an appendix because, although they are not directly cited in the body of the narrative, they help to understand it in its generality and serve as an interpretative roadmap for the data (previous tables numbers 5, 6 and 7).

The conclusion summarises the objectives we set out in the introduction. I believe that reducing its size, which is currently around 550 words, would mean lowering its quality.

The changes resulting from this and the other evaluations have been signalled in yellow.

The author.

Reviewer 4 Report

Comments and Suggestions for Authors

The subject of the article seems to me to be very interesting and contributes to our knowledge of the use of materials in the past. It's very informative, even though it's largely based on well-known sources. In scientific terms, I have nothing to criticise, but the way in which the content is presented leads me to make a few comments.

Firstly, the reference to the domestication of colour in the title and the development of the subject at the beginning of the introduction seem excessive to me, because the article, despite the historical background, is essentially limited to the presentation of procedures (for quality control and dyeing).

The article has a formal development, characterised by a development that is less focused on the main subject and with some prolixity that fits in with the practices of History, but less so with the narrower and more rigid conventions of the Sciences (this is a question that may or may not be relevant depending on the journal's perspective on its placement).

Related to this is a set of notes which, if not in their entirety, at least in large part are unnecessary, and the little information that is actually relevant should be inserted into the text itself, to the benefit of readers.

Some parts of the article, due to the context in which they appear, suggest at first reading that it is a development of the Portuguese case, but it is later verified by the reference that they correspond to general descriptions, with the data from the sources used only appearing later (for example, lines 443 and following). It becomes unnecessarily unclear to the reader.

The numbering of the tables doesn't follow the order in which they are mentioned in the text and the distribution of the tables throughout the article is strange (some in the text, others at the end). By the way, on line 647, "Table 3" should read "Table 12".

Table 2 mentions "vereadores", but in the text there is no reference to them, but to "vedores". Are they the same? Clarify.

In note 18, a comma is used instead of a dot as the decimal separator.

Finally, I have doubts about the appropriateness of the way the cited pages of the references are indicated in the text.

Author Response

I would like to thank you for your careful review and comments, which were very useful for the work.

With regard to the use of the concept of domestication, my idea was to emphasise the role played by textile dyeing technologies in human interaction with colours and the natural resources capable of reproducing them. It seems to me that a more extensive understanding of this concept, involving social and cultural relationships, could be interesting as a motto for other studies. The use of this word is primarily for narrative purposes.

As for the footnotes, we have reduced the number (from 44 previously to 36 now) and the graphic space occupied by the section. This reduction was made without diminishing the clarity of the ideas or arguments presented.

To improve the clarity of the arguments, the tables have been reorganised in the order in which they appear in the text. I created an appendix section with three tables which, although they are not cited throughout the text, serve as general support for the work and for understanding the data presented.

Specialised bibliography was used to compare and discuss the empirical data. This narrative resource is common in historiography.

'Vereadores' (members of the council's administration) and 'Vedores' (responsible for the council's economic supervision) are distinct offices. Throughout the text, in this and other terms, we have chosen to explain them in English using parentheses and brackets.

The situation in note 18 has been corrected.

As for page citations, we have followed the guidelines provided by the journal on its official website.

Changes resulting from this and other evaluations have been marked in yellow.

The author.

Reviewer 5 Report

Comments and Suggestions for Authors

Reviewer Recommendation and Comments for Manuscript ID: heritage-2839051 entitled Domesticating colour in the Early Modern Age: dyeing wool in black in Portugal for Heritage.

The manuscript entitled Domesticating colour in the Early Modern Age: dyeing wool in black in Portugal by Luís Gonçalves Ferreira is an interesting study in the area focus on the History of Black Clothing from methodologies of Cultural and Social History (Early Modern Age) in Portugal.

Both the main objective and the three secondary objectives have been successfully achieved. They have been adequately argued.  Therefore, I would recommend publishing this study.

Author Response

As no relevant changes have been mentioned, I would like to thank the reviewer for their interest in my manuscript and the work done to revise its content.